# Insulating Material Development for the Design of Standoff Insulators Fed by Hybrid Voltage

**DOI:** 10.3390/ma15155307

**Published:** 2022-08-02

**Authors:** Gian Carlo Montanari, Riddhi Ghosh, Robin Ramin, Debasish Nath

**Affiliations:** Center for Advanced Power Systems, Florida State University, Tallahassee, FL 32306, USA; riddhi.ju@gmail.com (R.G.); rramin@fsu.edu (R.R.); dnath@fsu.edu (D.N.)

**Keywords:** standoff insulators, bushings, dc and ac supply voltage, design optimization, hybrid supply, electrified transportation assets, reliability, finite element analysis, surface field and discharges, creepage and clearance

## Abstract

Innovative electrical assets are being developed in transmission and distribution, as well as in electrified transportation, from ships to aerospace. In general, power electronics have to master the whole power supply, being the driver of high specific power, low weight and volume components, in addition to enabling flexible and highly variable power flow. In these conditions, electrical and electronic insulation systems will have to withstand new types and levels of electric stresses, while still maintaining its reliability throughout its whole design life. This paper presents a study on the interrelation between insulating material properties and surface field of standoff insulators. The aim is mainly to provide indications on material properties which can be tailored to provide a robust, reliable and optimised insulator design that will hold for any type of electrical stress the insulation will have to withstand during operation. Specifically, we focus on ac and dc supply, including voltage transients, which could feed the same insulator depending on operation, according to a hybrid asset paradigm. The challenge is, indeed, to establish a pattern to material and insulation system design which takes into account the differences between the types of electrical stress profile and magnitude when insulators are supplied either in a dc or in ac, in order to infer which type of material characteristics would be more appropriate for the sake of life and reliability. The main contribution of this paper is to show that engineering the values of bulk and surface conductivity (which can be done selecting appropriate materials or modifying them, e.g., by nano-structuration) and modelling surface discharge inception would allow the electric field profile to be stabilised whatever the shape of the applied waveform. This will enable us to reach a reliability target that not only accounts for macroscopic phenomena, but also for the likelihood of extrinsic accelerated aging mechanism occurrence as partial discharges. In such a way, optimization of conditions to improve life, reliability, design and creepage and clearance characteristics can be achieved.

## 1. Introduction

Innovation in electrical assets has to deal with extreme optimization and exploitation of any electrical and electronic asset component. Drivers are high power density, low weight and volume, large power and broad energy dynamics, high efficiency and the capability to withstand changes in asset and mission. These requirements are met using power electronics, high frequency and ultrafast switch components. Hence, ac sinusoidal voltage is being replaced by modulated sinusoidal and dc voltage through, e.g., Pulse Width Modulation (PWM) technology. Hybrid voltage supply, providing the option of changing dc into ac modulated voltage when feasible and convenient, will be allowed. Indeed, there are more and more studies and applications of a hybrid approach to electrical grids [1,2,3,4,5], and this holds also for electrified transportation [6,7,8].

Since life performance and high reliability are also fundamental bricks of this architecture, maximum attention shall be paid to electrical insulation, which is often the prevailing source of failure of electrical and electronic asset components. A higher voltage and electrical field, higher frequency and operating temperature and the presence of fast repetitive voltage impulses will affect, and generally worsen, electrothermal-stress aging rate, and potentially cause dramatic life and reliability reduction with respect to design specifications [9,10,11]. A counteraction to face such threat is conservative and redundant insulation system designs, including, e.g., larger insulation thickness, broader creepage for standoff insulators and, in general, a longer distance between electrodes. However, this is not an option where power density, volume and weight are design constraints. Therefore, the proper answer is to investigate the impact of material properties and design on the new types of stresses and associated ageing mechanisms, model life and develop effective diagnostic tools that can allow Condition-Base Maintenance (CBM) to be carried out. This paper focuses on a standoff insulator design, investigating the surface electric field behavior from ac supply to dc steady state, considering also the effect of voltage transients and how they affect the electrical field distribution in insulation bulk and on its surface. Those material properties, e.g., conductivity, that can be modified in order to optimise design, minimizing gradients and variation in the electric field profile in the presence of load changes and type of electric supply (ac, dc, transients) are varied to highlight their impact on design. This can provide an indication of the extent of advantages that can be obtained in engineering such materials for hybrid supply, which adds up to existing published work or standard indications [12,13]. The innovation of this study is, however, not only in material engineering directives, but also in guiding towards optimised designs which allow insulation to perform reliably under both ac and dc supply, minimizing insulation thickness, creepage and clearance. Various values of bulk and surface conductivity are considered in order to provide a basis to insulating materials engineering, profiting of, e.g., different compounds or adding nanofiller. In this way, the maximum value and gradient of the surface field can be reduced and its variation with type of applied voltage hampered, with evident advantages of achieving optimised designs working reliably under both ac and dc supply and of minimizing creepage and clearance. Section 2 describes the simulation model and the basis equations to calculate the electric field for the reference spacer geometry used for the modelling, which relies upon Finite Element Analysis (FEA). Innovation here is to consider a thin surface layer of the material, which accounts for the different properties of bulk and surface conductivity of an insulation. This can affect profoundly the surface field calculation. Section 3 shows the results of the conduction current and space charge measurements (bulk and surface) on a resin (fiberglass) typically used for spacers to establish a solid basis for field calculation as a function of conductivity characteristics, which is the topic of Section 4. Section 4 also discusses the changes in field behaviour under ac, dc and voltage transients with material properties, and its impact on surface discharge likelihood, and Section 5 finally speculates on material and design optimization under hybrid voltage supply. As a final note, the presence of defects or surface contamination is not taken into account here, but this is a fundamental issue that could be fitted into the presented framework and can be developed in a next paper.

## 2. Electrical Field Modelling

The purpose of electric field calculations is to support risk evaluation of surface discharges and to optimise creepage and clearance, as well as insulation thickness, for ac and dc spacers. Hence, an object as that in Figure 1 in [14] was considered for modelling and simulation purposes. A novelty here is to separate surface and bulk insulation behaviour through considering a surface layer, used to simulate the different surface conductivity properties compared to bulk conductivity (and also to possibly account for contamination). Surface, even if perfectly clean and dry, can have conductivity values, also for homogeneous and isotropic materials, much higher (even orders of magnitude) than the bulk, due to the different microstructure [14,15]. The modelled surface has a thickness of 100 µm (a thinner layer does not change the simulation results noticeably [16]). The surface conductivity for the reference calculations is one order of magnitude higher than the bulk conductivity, and lower than that of air at room temperature [16]. The parameters for the solid materials used for the initial basis simulations are summarised in Table 1, being taken from the literature and previous authors’ work [14,15,16,17]. The values for air are derived from the FEA library (COMSOL v5.5), except for the electrical conductivity, which was approximated from [18]. The electrical conductivity of the spacer bulk and surface depends on the electric field and temperature, and can be modelled by Equation (1), where *σ*_0_ is the reference conductivity measured at reference field *E*_0_ and temperature *T*_0_, *α* and *β* are the temperature and field dependent coefficient [8,10]. It must be noted that Equation (1) is an approximation of the Arrhenius law, working, however, with good accuracy in the considered temperature range [14,19,20].
(1)$$\sigma \left(E,T\right)=\sigma_{0}e^{\alpha \left(T-T_{0}\right)+\beta \left|E-E_{0}\right|}$$

The coupled Maxwell’s equations are as follows:∇ · *J* = *Q*(2)
(3)$$J=\sigma E+\frac{\delta D}{\delta t}$$
*E* = −∇*V*(4)
*q* = −*k*∇*T*(5)
(6)$$\rho C_{p}\frac{\delta T}{\delta t}+\rho C_{p}u\;\cdot \;\nabla T=\nabla \;\cdot \;(-q)+Q_{e}$$
where *J* is the current density, *Q* the electric charge, *D* the displacement field, *V* is the voltage, *q* is the rate of heat flow, *k* is the thermal conductivity, *ρ* is the matter density, *C_p_* the heat capacity, *u* is the velocity vector and *Q_e_* is the external heat source.

The results of the electric field simulation on insulator bulk and surface are shown in Figure 1a,b, where the *y*-axis describes the distance between electrodes, starting at the bottom electrode (0 mm) and ending at the top electrode (30 mm). The *x*-axis accounts for bulk insulation thickness and surface distance, respectively. The *y*-component of the field, *E_y_*, represents either the orthogonal bulk field or the tangential surface field. The *x*-component of the field, *E_x_*, is the radial field.

As can be seen in Figure 1b (relevant to the model parameter values of Table 1), at room temperature, the surface ac and dc field distributions, which are driven by permittivity and conductivity, respectively, have a significantly different magnitude and profile. In particular, the maximum surface tangential dc field is 20% higher than the maximum ac field. This may have a non-negligible impact on the surface discharge risk and in creepage and clearance distances when going from ac to dc supply (Section 5). On the other hand, the bulk field is similar for ac and dc due to the uniform-field setup of two parallel metal electrodes (and the mild dependence of conductivity on field which the major cause of the profile deviation from dc to ac under isothermal conditions).

Figure 2 highlights that with the increasing temperature (65 °C) the dc surface field decreases its maximum value and its difference with the ac field becomes much lower, at least for the considered conductivity material parameters. This is due to the increase in surface conductivity, which becomes even larger than that of air [16]. The bulk profiles for ac and dc are almost coincident. Hence, increasing the operating temperature uniformly in the whole insulation may lead to an overall improvement in field distribution from ac to dc, at least with the material characteristics from Table 1.

However, when a thermal gradient is considered (40 °C in in Figure 3, where the top electrode is hotter than the bottom electrode), the ac and dc field profiles in the bulk and on the surface differ significantly. This is due to the temperature coefficient value, Equation (1). The field in the bulk is shifted towards the bottom electrode and its maximum value increases to approximately 260% compared to the ac maximum value. On the surface, a significant change in the tangential field is observed, with the maximum tangential field value increased by 46% compared to ac, which is due to the decreasing conductivity from the top to bottom electrode as a consequence of the temperature gradient and *α* value. This significant local increase in maximum field amplitude may lead to the higher likelihood of partial discharges (PD) and it may also have a non-negligible impact on the surface discharge risk and in creepage and clearance distances (Section 5).

## 3. Conduction Current and Space Charge Measurements

In order to derive parameter values that are related to dc field distribution, i.e., those relevant to Equation (1), for other materials that can be used in standoff insulators, and to compare the field profiles with those provided and reported in Equation (1), bulk (volume) conductivity measurements were performed at different electric fields (2.5 kV/mm to 15 kV/mm) and temperatures (25 °C to 60 °C). The tested specimens belonged to two different types of materials, i.e., a loaded epoxy resin and a fiberglass, indicated in the following as materials #2 and #3 (while material #1 is that parametrised in Table 1). The mean specimen thickness was 0.87 and 1.05 mm for fiberglass and epoxy. Surface conductivity measurements were also performed at 25 °C and 50 °C and fields 0.05 to 0.2 kV/mm. Measurements were repeated from two to three times to check for repeatability. $\sigma_{0},\;$*α* and *β*, Equation (1), were then estimated for materials #2 and #3.

As shown by Table 2 and Figure 4, the bulk conductivity values for material #1 and #2 are similar, while that for material #3 (fiberglass) is three orders of magnitude higher (due to the significant anisotropy caused by the fibre orientation). Regarding surface conductivity, the values for the three materials are closer, within one order of magnitude difference. *α* is the highest for material #1 (surface and bulk) and the lowest for material #2 (bulk). *β* varies strongly, not only for different materials, but also between the bulk and surface of the same material. The *β*-values in the bulk are two orders of magnitude lower than on the surface, which may lead to large conductivity difference between bulk and surface, even if *α* is similar when field gradients are present. Noticeably, the surface field of material #2 (fiberglass) will be driven by the huge *β*-value of over 30 mm/kV. This can be addressed again to the anisotropy of the material, where fibres can constitute a preferable path for the conduction process. Eventually, for both materials #2 and #3, surface conductivity is higher than air conductivity also at a low temperature (contrarily to material #1).

The experiments reported here and the literature data further highlight that the values of bulk and surface conductivity can vary in orders of magnitude depending on the type of material, manufacturing process, temperature and design field. Additionally, it can be speculated that additives, reinforcing fibres or particles (including nanofillers), can play a fundamental role in modifying, even significantly, conductivity and its dependence on temperature and field. Hence, it can be figured out that materials, manufacturing and design procedures could be engineered by achieving conductivity parameter values for bulk and surface that allow the electric field profile to be stabilised and minimised in terms of the maximum field, when supplying the insulation system from ac to dc. This is the reason for the sensitivity analysis carried out in the next Section.

As a note, conductivity values in Figure 4 and in Table 2 correspond to the quasi-steady state values obtained from charging current characteristics (measured at different fields and temperatures): as can be seen, they fit quite well to Equation (1).

As a last note: space charge measurements were performed using a pulsed electroacoustic device (PEA) up to a poling field of 10 kV/mm, without observing significant space charge build up inside insulation. Regarding surface, most likely surface discharges, if any, could then be the major source of space charge accumulation, which is another motivation for the modelling part described in Section 5.

## 4. Bulk and Surface Field Distribution under AC, DC and Voltage Transients: Sensitivity to Conductivity Parameters

The electric field simulations in steady state dc were carried out considering a time *>* 5*τ_d_*, being *τ_d_* the dielectric time constant that rules electric field build up in insulation after a voltage step [10,14]. The dielectric time constant can be calculated, for homogeneous materials, as [21,22]:*τ*_*d*_ = *ε*_0_*ε*_*rb*_/[*σ*_*b*_(*T*(*x*, *y*, *t*), *E*(*x*, *y*, *t*))](7)
where *ε*_0_*ε_rb_* is the permittivity of the dielectric material and *σ_b_* its electrical conductivity at a selected temperature and field. With the values considered for Figure 2 and Table 1, *τ_d_* ranges between ~1200 min and ~22 min at isothermal 25 °C and 65 °C, respectively. In the presence of a thermal gradient, there is a distribution of time constants inside the material, depending on local temperature. Hence, the maximum time constant is defined by the conductivity at the coldest area in the test/simulation setup. As such, the time constants can be orders of magnitude longer than the voltage transient time, as it takes 10 s for the voltage to rise and reach a steady state value of 10 kV (used for Figure 1 and all following simulations). The behaviour of bulk and surface field in the y-direction is displayed in Figure 5, Figure 6 and Figure 7, going from ac sinusoidal to dc-steady state voltage, for the data of Table 2.

This next section focuses on the bulk orthogonal field (which might be associated with internal partial discharges, in case of defects) and the tangential surface field, which is related to surface partial discharges and to design creepage. In addition to dc, steady state and ac sinusoidal, three times during the field transient, after voltage step application, are considered, i.e., 0.1*τ_d_*, *τ_d_* and 5*τ_d_*. Since the bulk material is assumed to be homogenous and the field is mostly uniform, see Figure 5, Figure 6 and Figure 7, the transient under isothermal conditions does not influence the field distribution noticeably (note that the *x*-axis scale in Figure 5, Figure 6 and Figure 7 is reduced compared to Figure 1, for the sake of better viewing the transient behavior). Regarding the surface field behaviour, it can be seen that for the selected values of conductivity (which can be lower or higher than that of air), the field magnitude and profile change significantly from ac and initial energization time to steady state dc. A contribution is given also by the presence of a triple point at the electrode, involving air and insulation surface, and the relevant field gradient. As expected, the ac profile coincides with the initial transient field profile, since at the beginning of the voltage transient (after the step voltage), the field is mostly driven by permittivity, as in ac, while approaching steady state it is conductivity to rule the electric field distribution. After ~5*τ_d_*, the transient and steady state dc fields practically coincide. Therefore, it can be speculated that the field in dc systems after a voltage transient (as energization) behaves mostly like an ac field for times << *τ_d_* and a dc field when approaching *τ_d_*. It is noteworthy that the transient time calculations from Equation (7) differ slightly from those obtained from simulations. This can be explained by conductivity dependence on electric field and temperature, as seen in Equation (1).

The maximum field values in dc are larger than in ac for material #1 and #3, considering both the bulk and surface tangential field (even if changes in bulk are negligible for isothermal conditions). Material #2 has the lowest maximum electric field for both bulk and surface, and it shows the lowest difference in maximum electric field between dc and ac conditions (−6%, while for #1 and #3 it is larger than 15%). The orthogonal component of surface field (not reported here) is lower than the tangential component for all materials. The surface field can change drastically with surface conductivity modification in relation to the air conductivity value. The latter, as shown in Figure 8a (taken from [18]), can vary with ambient humidity, but it seems that relative humidity variations do not cause too much of a large effect on the electric field, as highlighted in Figure 8b. Hence, a value of *σ* for air as in Table 1 seems to be justified for the calculations reported here.

As the bulk material is assumed to be homogenous and the orthogonal field is mostly uniform, the transient under isothermal condition does not influence the field distribution noticeably. Regarding the surface, it can be seen that for the selected values of conductivity, lower or higher than that of air, field magnitude and profile change significantly from ac and initial energization time to steady state dc (due to the presence of the triple point at the electrode and the relevant field gradient).

Examples of electric field distributions with thermal gradients are shown in Figure 9 and Figure 10, where the temperature gradient is 20 °C (from 45 °C at the upper electrode of Figure 1a in [14] to room temperature), which can be compared with Figure 1 derived at room temperature (isothermal). They are relevant to the three considered materials and the conductivity coefficient values of Table 2. According to Equation (1), conductivity is increased in the hotter region and this results in a significant decrease in the electric field, while the opposite occurs in the colder areas. As a general consideration, varying the values of *α* and *β* can modify significantly the dc field distribution and its distance from the ac one, but in a complex way. For example, *β* for material #2 is large, Table 2, but the difference between the ac and dc tangential fields is the lowest, Figure 10.

Referring to the tangential field, material #2 has the best field distribution for isothermal and thermal gradients. However, the orthogonal field for material #2 is the largest compared the other materials, although #2 keeps showing the smallest difference between ac and dc regarding the orthogonal, radial and tangential fields (Figure 9 and Figure 10). The large value of *α* causes, on the other hand, a strong deviation between cold and warm areas of Material #1.

Summarising, an insulator fed by dc voltage operates in mixed conditions during voltage transients. Surface field distribution and magnitude can vary considerably from ac to transient conditions to dc steady state, depending on the values of bulk and surface conductivity and their dependence on field and temperature, Equation (1). This is why the above sensitivity analysis, pointing out the impact of conductivity coefficient values on ac, dc and transient field profile modification, can provide useful hints for the design and material choice and engineering. From this analysis, it was determined that the minimization of the shift from ac to dc field profile, including transient, and the minimization of the maximum field might be driven by choosing low values of *α*, high *β*, and *σ*_0._

## 5. Discussion and Impact on Insulation Material Engineering and Reliability

The distance between electrodes in Figure 1a in [14], used for the simulations in this paper, is slightly lower than that specified for the clearance of a clean insulator under ac voltage. A criterion to establish a creepage value holding under hybrid condition (and for uniform surface properties, including pollution) is to refer to an estimation of mean electric field for surface discharge inception, which may hold when the field is quite uniform on an insulation surface. Reliable operation of the insulator, on the other hand, should not experience any surface partial discharge, both in ac and dc, because this would affect the insulator’s reliability and generate a condition for destructive discharges. A new expression that can help to estimate the condition for surface PD inception (which is, however, deterministic and strongly approximated) can be derived from [23]:(8)$$\frac{E_{0i}}{E_{cr}}=F[(\mathrm{pl}),\;\mathrm{defect}\;\mathrm{geometry},\;\mathrm{material}\;\mathrm{characteristics},\;\mathrm{gas}/\mathrm{polymer}\;\mathrm{interface}]$$
where *p* is pressure, *l* distance, *E*_0*i*_ is the surface discharge inception field and
*E*_*cr*_ = *E*_*ch*_*/γ*(9)
being *E_c_**_h_* the field established along the streamer channel during discharge propagation and *γ* a dimensionless factor that depends on the gas or gas/surface combination and on the streamer probability. It is claimed in [23], Table 2, that the contribution of the polymer surface to gas ionization is negligible, but the large difference between surface discharge and discharges in a gas-filled embedded cavity indicates clearly that surface/interface properties play a significant role in modelling the surface partial discharge inception. Hence, the values of the parameter that define *F* in Equation (8) can be referred to the gas surrounding insulator (air in our case), as well as to the type of defect and its geometry and the material characteristics, specifically relative to permittivity and conductivity (for ac and dc), see [24].

This furthermore highlights that obtaining accurate calculations/simulations of the electric field profile on the insulator surface, focusing on the field in the direction of the streamer development; that is, the tangential field, is of paramount importance to avoid the inception of surface PD, as well as, in some cases (almost uniform surface tangential field), creepage and clearance optimization. Additionally, since the PD inception field does not depend (neglecting roughly the stochastic aspects associated with PD inception) on supply voltage waveform, but on its maximum value, keeping the surface field profile and maximum field as constant as possible between ac and dc supply will allow an optimised, rather than conservative, design. This, in turn, should guarantee the specified reliability based on, e.g., the ac insulator design, on which there is much more experience.

Regarding bulk field, this has to do with the life model and intrinsic electrothermal ageing. Intrinsic ageing is that caused by the stresses accounted for during the design procedure [24], which is conducted based, in general, on an electrothermal life model as:(10)LD=ff0t0( EDES0 )−n
where *E_D_* and *L_D_* are design field and life at a selected temperature, *E*_0_ is the reference electric field (generally estimated by short-term accelerated life tests), *t*_0_ is the failure time at *E* = *E*_0_, *n* is the voltage endurance coefficient, which is the inverse of the life-line slope in log-log coordinates, *f* and *f*_0_ are supply voltage (modulation) frequency and reference frequency, respectively (*f/f*_0_ = 1 under DC) [19]. Accelerated life tests are used, in general, to estimate the model parameters and are based on the orthogonal bulk electric field value, which must be equal to or lower than *E_D_*. Keeping in mind *E_D_* and the electric field profile, the insulation thickness can be calculated for the operating voltage. The higher the orthogonal field, the thinner insulation. However, at the design stage, a conservative approach may take into account the risk of partial discharge, PD, incepting in cavities formed during manufacturing, which can cause accelerated extrinsic ageing and failure [25]. The lower the bulk design field, the lower the probability of PD inception and, in case of PD occurrence, discharge pulse magnitude. Besides intrinsic ageing, which is ruled by Equation (10), an insulation system can experience extrinsic ageing; that is, the ageing processes occurs due to unpredictable factors such as space charge, partial discharges and hot spots.

To summarise, the design and manufacturing of materials for stand-off insulators, which will go towards the direction of higher bulk conductivity, would flatten the dc field profile and reduce the magnitude of the maximum field, as shown in Figure 11. Further positive effects on field control can be obtained by decreasing the dependence of conductivity on temperature (*α*) and increasing (but only slightly) it on the field (*β*) in order to smooth the dc field profile, with maximum amplitude lower than under ac also in the presence of thermal gradient Figure 11. The above results indicate the method to follow in the engineering of insulating materials. Indeed, it can be speculated that the dc tangential surface fields, as large as those in Figure 11; that is, using low conductivity materials, are quite far from surface discharge risk conditions, both in ac and in dc (under the assumption that discharge is triggered by the maximum field value, while the PD repetition rate depends on the type of supply voltage shape [11,26]). Likewise, the orthogonal bulk field is very low, well below the potential design field for such types of insulators (which could go up to some kV/mm). Thus, there is apparently room for spacer dimension optimization, unless contamination level criteria and concern would prevail, making a very conservative design the most appropriate choice.

These results indicate that modelling the critical situations for reliability (as, e.g., partial discharge inception field or voltage) and engineering insulating materials, e.g., by the addition of surface treatment/surface and bulk nanofillers, could allow an optimum design to be achieved that can enable operation below the surface partial discharge inception voltage with, possibly, reduced creepage and clearance dimensions. This can provide the specified reliability for a long time, even for insulators that can be used both under ac and dc voltage waveforms. Properly managing the choice of materials seems, therefore, a key issue to reach the needed high reliability level in electrical assets used in electrified transportation, irrespective of the type of supply voltage waveform. As a final note, increasing the ac supply frequency to some kHz, as for the most recent PWM supply, would not change the ac field distribution noticeably, as the ac and high-frequency ac field profiles totally overlapped.

The operation conditions are 10 kV at 25 °C and the model parameter values were taken from Table 1.

## 6. Conclusions

It is evident from the results presented and discussed in this paper that an insulation system where surface plays a fundamental role to safely and reliably separate HV from LV conductive parts should be designed in a different way, whether it has to be used under ac or dc voltage. Clearance and creepage in dc might correspond to a longer distance between electrodes than in ac, especially if the thermal gradient is significant, while during voltage transients a mix up of both conditions, that can last minutes to hours, has to be expected. Such achievements indicate that the diffusion of the concept of hybrid voltage supply in electrified transportation assets would be problematic from an insulation reliability point of view, if appropriate countermeasures are not taken at the insulator design stage. A solution described here is to work on surface partial discharge modelling and on insulating material engineering, which could help in achieving an optimised design regarding dimensions and reliability. Indeed, it has been shown above that having bulk and surface conductance which can flatten the dc field profile decreases the maximum field at values very close to those expected in ac (for both tangential and orthogonal components, but mainly for the former). In this way, the type of voltage waveform would not affect insulator life and reliability (just its peak value), appropriately taking into account the frequency effect and the possible presence of defects. In other words, an insulator designed for sinusoidal ac could also work properly in dc conditions and, most likely, for modulated ac (referring to peak voltage and modulation frequency).

This opens the door to material design and optimization, tailoring properties so that hybrid supply will not cause dramatic aging acceleration and life reduction.

To summarise, even if design optimization is still a challenge, the modelling provided in this work may at least help to avoid simply resorting to worse conditions and the over-design of a standoff insulator. However, more refined simulations are needed, especially for in regard to surface contamination.

## Figures and Tables

**Figure 1 materials-15-05307-f001:**
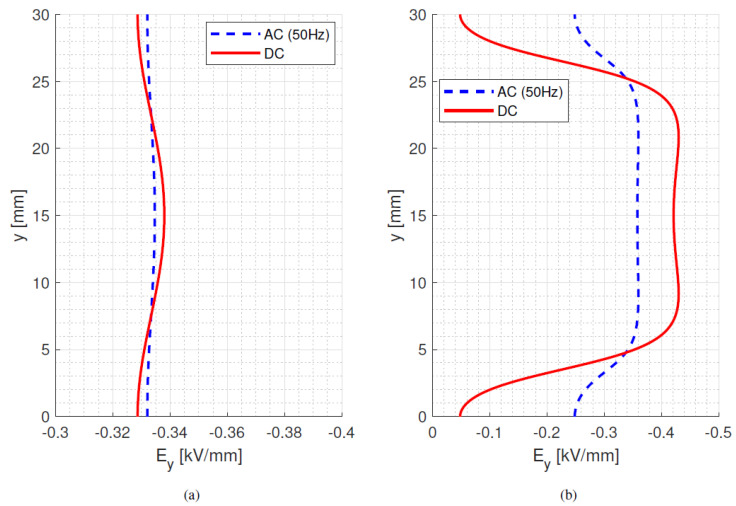
Bulk (**a**) and surface (**b**) ac and dc steady-state field for 10 kV at 25 °C. Model parameter values from Table 1.

**Figure 2 materials-15-05307-f002:**
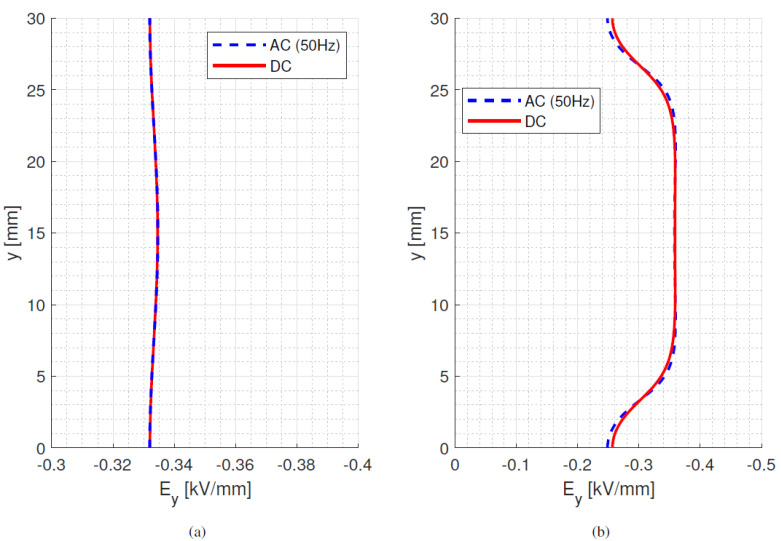
Bulk (**a**) and surface (**b**) ac and dc steady-state field for 10 kV at 65 °C. Model parameter values from Table 1.

**Figure 3 materials-15-05307-f003:**
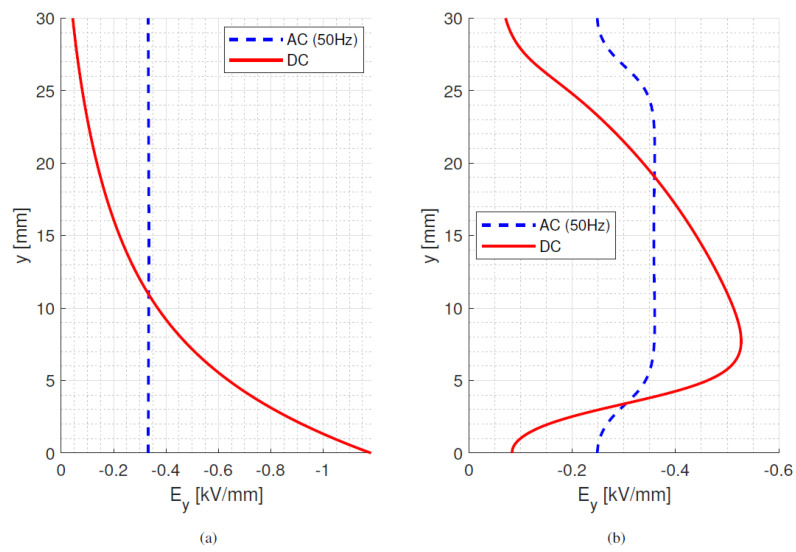
Bulk (**a**) and tangential surface (**b**) ac and dc steady-state fields at 10 kV with 40 °C temperature gradient, from 25 to 65 °C. Model parameter values from Table 1.

**Figure 4 materials-15-05307-f004:**
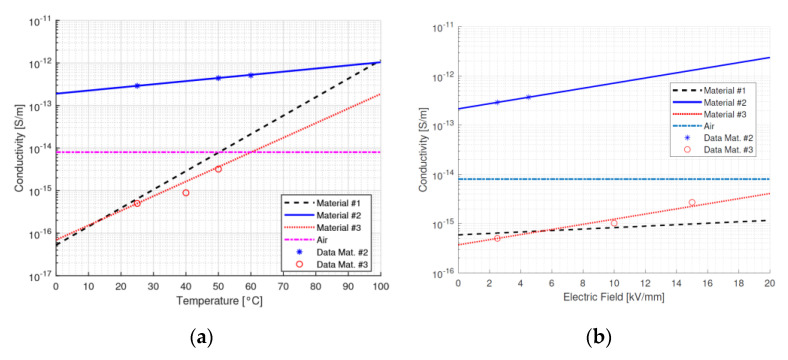
Bulk conductivity for materials #1 to #3 and air as a function of (**a**) temperature at field E = 2.5 kV/mm and (**b**) electric field at T = 25 °C.

**Figure 5 materials-15-05307-f005:**
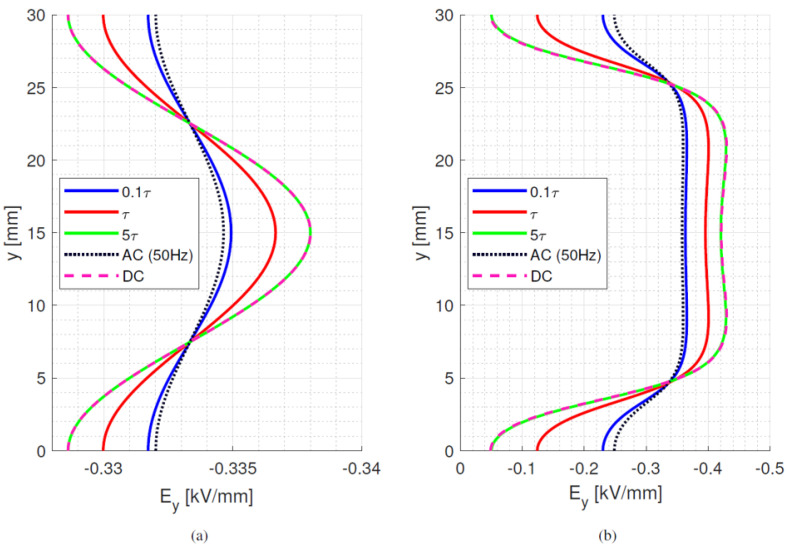
(**a**) bulk orthogonal and (**b**) surface tangential field at different times after voltage step application till steady state dc and sinusoidal ac (50 Hz) for material #1. Time constant *τ_d_* = 70 min. Isothermal condition at 25 °C. Test object of Figure 1a.

**Figure 6 materials-15-05307-f006:**
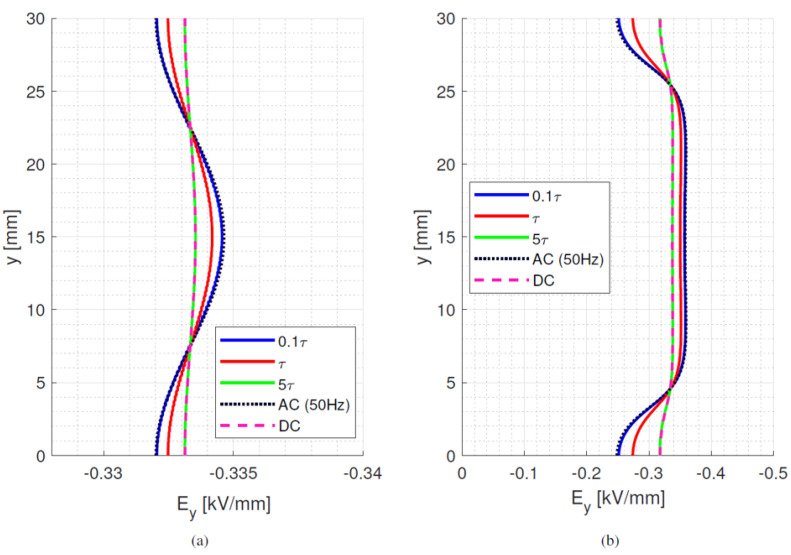
(**a**) bulk orthogonal and (**b**) surface tangential field at different times after voltage step application till steady state dc and sinusoidal ac (50 Hz) for material #2. Time constant *τ_d_* = 1.6 min. Isothermal condition at 25 °C. Test object of Figure 1a.

**Figure 7 materials-15-05307-f007:**
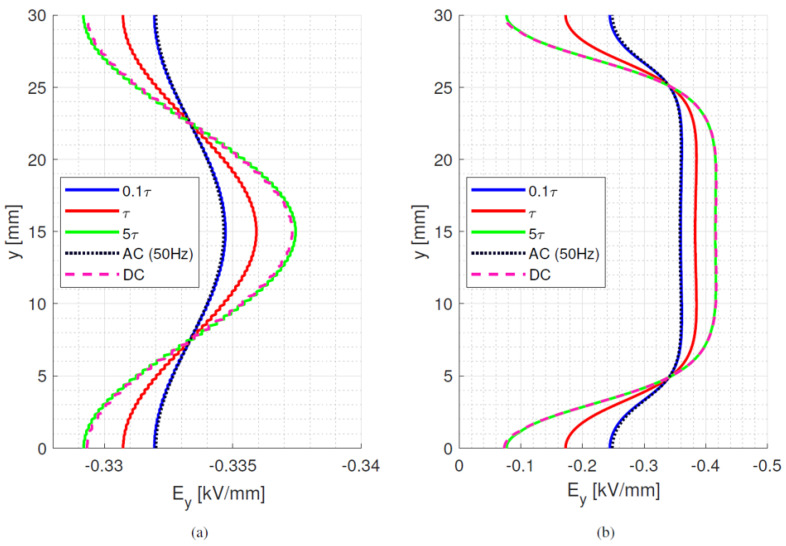
(**a**) bulk orthogonal and (**b**) surface tangential field at different times after voltage step application till steady state dc and sinusoidal ac (50 Hz) for material #3. Time constant *τ_d_* = 20 min. Isothermal condition at 25 °C. Test object of Figure 1a.

**Figure 8 materials-15-05307-f008:**
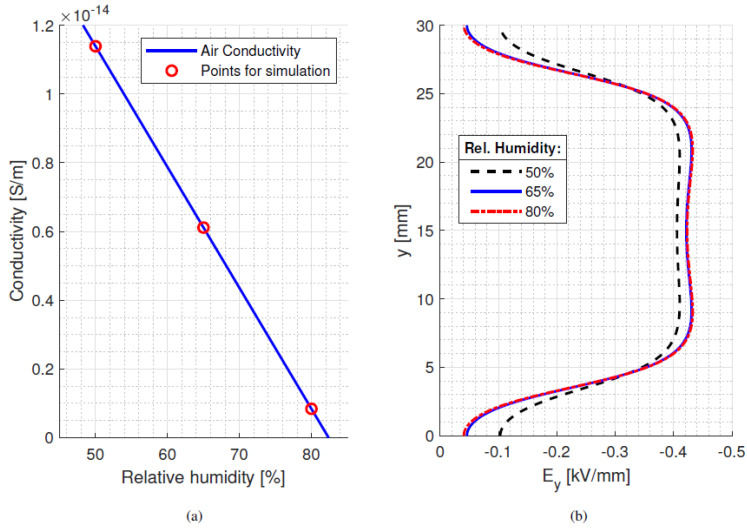
(**a**) Values of air conductivity at 25 °C as a function of relative humidity [18], and (**b**) impact on surface tangential field (considering the parameters of Table 1).

**Figure 9 materials-15-05307-f009:**
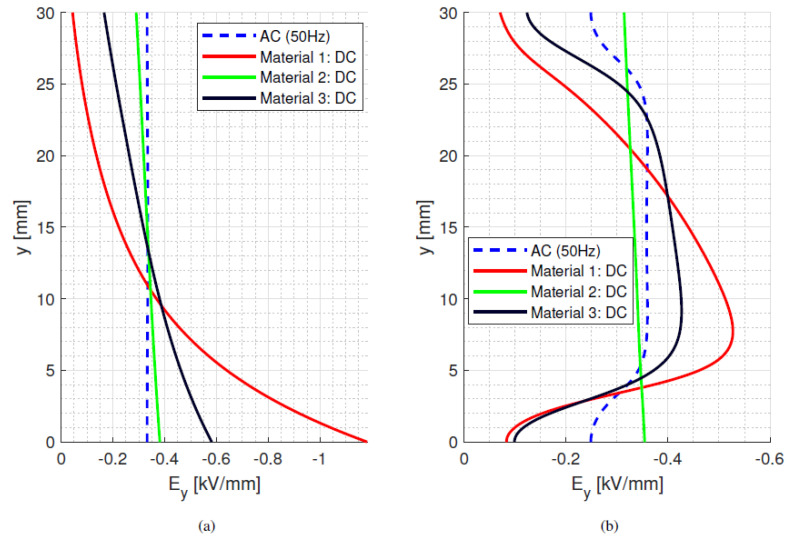
(**a**) Orthogonal bulk and (**b**) surface tangential field (Ey) for ac and dc steady state, with a temperature gradient of 20 °C (high-potential electrode warm, Figure 1a). Material characteristics of Table 2.

**Figure 10 materials-15-05307-f010:**
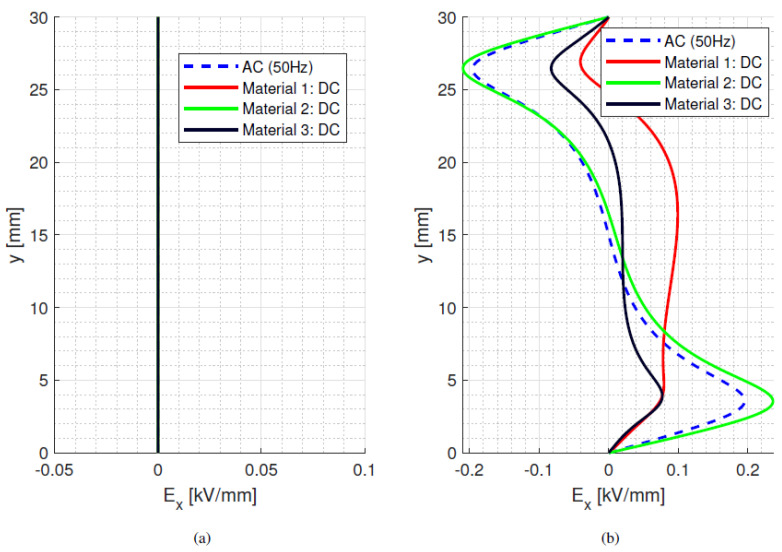
(**a**) Radial bulk and (**b**) orthogonal surface field (Ex) for ac and dc steady state, with a temperature gradient of 20 °C (high-potential electrode warm, Figure 1a). Material characteristics of Table 2.

**Figure 11 materials-15-05307-f011:**
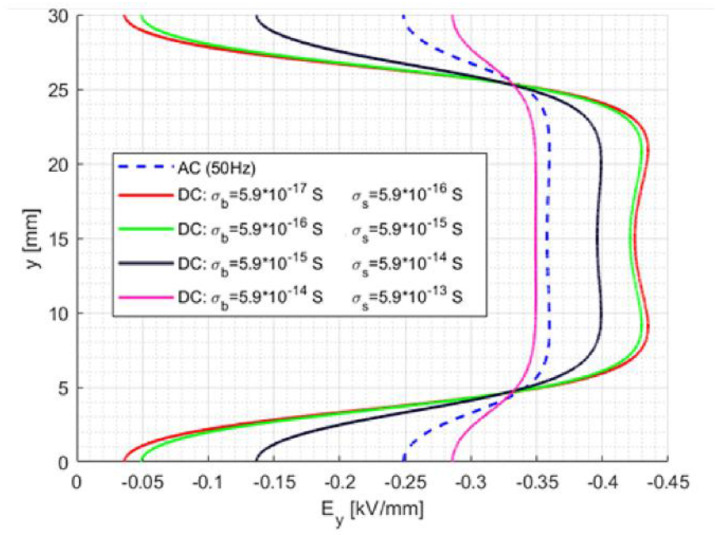
Tangential surface field for ac and for different conductivity values in dc steady-state.

**Table 1 materials-15-05307-t001:** Parameter of the insulator and air for the reference material.

Material Parameter	Insulator	Insulator	Air
	Bulk	Surface	
Electrical conductivity *σ*_0_	5.9 × 10^−16^ S/m	5.9 × 10^−15^ S	8 × 10^−15^
Temperature dependent coefficient *α* [1/K]	0.1	0.1	-
Electric field dependent coefficient *β* [mm/kV]	0.034	0.034	-
Reference electric field *E*_0_ [V/m]	0	0	-
Reference temperature *T*_0_ [°C]	25	25	-
Relative permittivity ε*_r_*	4.8	4.8	1
Thermal conductivity *λ* [W/(mK)]	0.274	0.274	*λ* (*T*)
Density *ρ* [kg/m^3^]	1800	1800	*ρ* (*P**,**T*)
Heat capacity *C_p_* [J/(kgK)]	1900	1900	*C_p_* (*T*)

**Table 2 materials-15-05307-t002:** Material parameter for different insulator materials @ 25 °C.

Material	*σ* _0_	*α* [1/K]	*β* [mm/kV]	*E*_0_ [V/m]
#1 Table 1 material Bulk	5.9 × 10^−16^ S/m	0.100	0.034	0.0
#1 Table 1 material Surface	5.9 × 10^−15^ S	0.100	0.034	0.0
#2 Fiberglass Bulk	2.9 × 10^−13^ S/m	0.017	0.120	2.5
#2 Fiberglass Surface	8.2 × 10^−14^ S	0.063	30.331	0.1
#3 Epoxy Bulk	5 × 10^−16^ S/m	0.079	0.084	2.5
#3 Epoxy Surface	2.9 × 10^−14^ S	0.07	2.977	0.1

## Data Availability

Not applicable.

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
