# Peer review of "Insulating Material Development for the Design of Standoff Insulators Fed by Hybrid Voltage"

_materials, 2022, doi:10.3390/ma15155307_

Round 1
Reviewer 1 Report
As the network operation is changing considering AC and DC power supply condition, added with PWM based power supply makes it more important to consider the impact of the new types of stresses on the insulation material. This paper addresses an important topic considering the behaviour of insulation material because of these new types of power supply conditions.
The paper is well organized, describing the significance of this research under current and future grid scenario. In-depth study is presented under ac, dc and voltage transients with material properties, and its impact on surface discharge likelihood using Finite Element Analysis simulation environment. The paper provides a valuable perspective in exploration robust material recopies for power components for extended withstanding capabilities.
However, it is important if the authors can describe in the paper about: considering PWM based power supply and its impact on the insulation degradation, one argument is that during conversion from DC to AC, there is a filtering process, which can reduce the impact of the higher dv/dt stresses. In this case, how important is to consider the impact of PWM based voltage transients on the aging of insulation material.
Reviewer 2 Report
The authors presented the results which are good. This paper looks interesting but some modifications are required.
1. Some typos and grammatical mistakes are there in the paper like - line no 20 (" It is shown that engineering the values of 20 bulk and surface conductivity.......").
2. The novelty of the paper has to be highlighted in the abstract. And kindly revise the abstract.
3. Authors are suggested to incorporate the objectives of the paper in the introduction section.
4. Bulk references of more than 2 have to be avoided and changed according to the format.
5. Font of fig.1(a) is not appropriate according to the format of the journal. quality of fog 1(b) is not according to the journal format (300ppi and above).
6. Font of all the figures should be according to the journal format.
7. Align all the equation numbers correctly.
8. Authors are requested to add a comparison table for the material used with recent literature.
Reviewer 3 Report
Understanding the electric field distributions in bulk and at tangential surface of standoff insulators is an important issue for the design and optimization of insulating structures. System simulations are carried out by G. C. Montanari et al., as well as some useful guidelines given. This work is recommended for publication after minor revisions.
1. Key parameters of the surface and bulk conductivities dominate the simulation results, thus, the selection of such parameters needs to be careful. The main concern is that whether the surface and bulk conductivities assigned to Material 1 in Table 1 are universal for solid insulators, and if yes, how come the larger varied performances of Material 2 and 3 being considered in this work. Moreover, the authors should indicate how serious the material will influence the general electric field distributions in conclusions.
2. In 428, the authors stated that “e.g. by addition of surface treatment/surface and bulk nanofillers, could allow an optimum design to be achieved that can enable operation below the surface partial discharge inception voltage with, possibly, reduced creepage and clearance dimensions”. It needs further detailed explanations or proper citations to support this issue.
3. The authors mention in line 437 that “being the ac and high-frequency ac field profiles totally overlapped”, but it is not directly supported by the fitting results, and needs further explanations.
4. Statement related to sample parameters needs to be corrected in page 7, line199-203, “As shown by Table II and Fig. 5, the bulk conductivity values for material #1 and #2 are similar, while that for material #3 (fiberglass) is three orders of magnitude higher (due to the significant anisotropy caused by the fiber orientation)”, should be “the bulk conductivity values for material #1 and #3 are similar, while that for material #2 (fiberglass) is…”.
5. The format at the paragraph header needs to be uniform, such as line 312, 382, and 392.
Round 2
Reviewer 2 Report
Accepted in the current revised manuscript